# Nonadiabatic Surface Hopping Dynamics of Photocatalytic Water Splitting Process with Heptazine–(H_2_O)_4_ Chromophore

**DOI:** 10.3390/ijms26104549

**Published:** 2025-05-09

**Authors:** Xiaojuan Pang, Chenghao Yang, Ningbo Zhang, Chenwei Jiang

**Affiliations:** 1School of Materials and Physics, China University of Mining & Technology, Xuzhou 221116, China; ts24180016a31@cumt.edu.cn; 2School of Mines, China University of Mining & Technology, Xuzhou 221116, China; znbcumt@126.com; 3Ministry of Education Key Laboratory for Nonequilibrium Synthesis and Modulation of Condensed Matter, Xi’an 710049, China; 4Shaanxi Province Key Laboratory of Quantum Information and Quantum Optoelectronic Devices, Xi’an 710049, China; 5School of Physics, Xi’an Jiaotong University, Xi’an 710049, China

**Keywords:** water splitting, carbon nitride materials, electron-driven proton transfer

## Abstract

Recent research on the use of heptazine-based polymeric carbon nitride materials as potential photocatalysts for hydrogen evolution has made significant progress. However, the impact of the water cluster’s size on the time-dependent photochemical mechanisms during the water splitting process of heptazine–water clusters remains largely unexplored. Here, we present a Landau–Zener trajectory surface hopping dynamics calculation for heptazine–(H_2_O)_4_ clusters at the ADC(2) level. The electron-driven proton transfer (EDPT) mechanism reaction from water to hydrogen-bonded heptazine–water clusters was confirmed using this method, yielding a heptazinyl radical and an OH biradical as products. The calculated quantum yield of the EDPT for the heptazine–(H_2_O)_4_ complex was 6.5%, which was slightly lower than that of the heptazine–H_2_O complex (9%), suggesting that increasing the water cluster size does not significantly enhance the efficiency of hydrogen transfer. Interestingly, our results show that the de-excitation of the heptazine–water complex from the excited state to the ground state via the EDPT process follows both fast and slow decay modes, which govern population relaxation and facilitate the photochemical water splitting reaction. This newly identified differential decay behavior offers valuable insights that could help deepen our understanding of the EDPT process, potentially improving the efficiency of water splitting under sunlight.

## 1. Introduction

A detailed understanding of the factors that drive water splitting under sunlight is central to an efficient EDPT reaction in hydrogen-bonded complexes [1]. The photoinduced water splitting paradigm outlined in this paper, originally proposed by Sobolewski and Domcke [2], differs from the previous photoelectrochemical [3,4,5] and artificial photosynthesis approach [6,7,8] by effectively avoiding the substantial charge recombination losses and enhancing the separation rate over long lengths and time scales, as the reaction occurs on the femtosecond timescale [9]. The hydrogen atom of the water, which is strongly hydrogen-bonded to a specific chromophore catalyst, can be oxidized efficiently to produce clean hydrogen (H_2_) energy through a biphoton absorption mechanism [9,10]. First, the hydrogen-bonded chromophore–water complex is excited to a higher energy state by absorbing the first photon, followed by hydrogen transfer from the hydrogen-bonded water to the chromophore through the EDPT reaction, producing biradicals containing hydroxyl and chromophore component radicals. Furthermore, the reduced chromophore radical is subsequently oxidized upon absorbing a second photon, thereby regenerating the catalyst and releasing a hydrogen atom in this step. Two hydrogen atoms are expected to combine spontaneously in the dark, ultimately generating molecular hydrogen (H_2_). As an alternative photocatalyst, graphitic carbon nitride—particularly the polymer C_6_N_9_H_3_, which can easily be synthesized from precursors [11]—has emerged as a promising material in the field of solar water splitting and has attracted widespread attention in recent decades [12,13,14,15,16,17,18,19,20].

One of the simplest graphitic carbon nitrides, an aromatic photobase named pyridine, was theoretically calculated by Liu et al. using the ADC(2) interpolation method and shown to absorb a hydrogen atom from water, forming a pyridinyl–hydroxyl biradical pair [10,21]. These predicted outcomes were also observed by Esteves-López et al. in molecular beam experiments, where it was found that photoexcited pyridine at 255 nm can abstract a hydrogen atom from a water molecule in pyridine–water clusters containing at least four water molecules [22]. Through comparative electronic structure calculations using the ADC(2) method, we have provided a plausible explanation for this, which is that the presence of the ^1^ππ* minimum substantially lowers the hydrogen transfer barrier height [23]. Along with trajectory surface hopping dynamics, an electron-driven proton transfer process from water to pyridine was confirmed in a simulation [24].

Recently, the single unit of hydrogen-terminated heptazine, C_3_N_4_, as the simplest unit structure of C_6_N_9_H_3_ in a photocatalyst field, enabled a series of achievements in the field of solar water splitting. It was also demonstrated by Ehrmaier et al. [19] through theoretical interpolation that heptazine plays a crucial role as a photocatalyst, absorbing a hydrogen atom from hydrogen-bonded water, with the product of the OH radicals being successfully detected experimentally [25]. Very recently, the molecular-level process of the EDPT reaction for a heptazine–water cluster was revealed by Ullah et al. [26] and Huang et al. [27] using the TDDFT and ADC(2) methods, respectively, at the excited state nonadiabatic dynamics simulation level. An average lifetime of tens of femtoseconds was confirmed for the EDPT process from water to a heptazine chromophore [26], and detailed quantitative insights into the reaction dynamics mechanism were obtained [27]. Interestingly, using relatively accurate methods such as the ADC(2), CC2, EOM-CCSD, and CASPT2 electronic structure calculation method, Ehrmaier et al. discovered that stable closed shell organic heptazine and polymeric carbon nitrides exhibited an extremely robust inverted S1/T1 energy gap, which was not affected by the substitution or oligomerization of its units. This feature shows a unique advantage for photocatalysis in the field of water splitting, with profound implications for the water splitting process [28]. The same inverted S1/T1 system was also confirmed and led to higher quantum efficiencies when using 2,5,8-tris(4-fluoro-3-methylphenyl)-1,3,4,6,7,9,9b-heptaazaphenalene (HAP-3MF) as a light-emitting diode in a study by Sobolewski et al. [29]. Lau et al. introduced a new system for storing sunlight photons, based on a cyanamide-functionalized polymeric network of heptazine units, which can efficiently absorb electrons, forming reductive radicals with long lifetimes exceeding the diurnal cycle, and emit these absorbed electrons to produce H_2_; in a dark environment [30]. Additionally, 3D-ordered closely packed fcc g-C_3_N_4_ nanosphere arrays were synthesized using a two-step nanocasting method and shown to significantly enhance water adsorption, accelerating the water splitting reaction and producing clean H_2_ [31]. Recently, ultrafast excited state dynamics were studied by Huang et al. and Ullah et al. using the ADC(2) [27] and TDDFT methods [26,32], respectively, confirming the EDPT reaction and providing a detailed mechanism. A real-time TDDFT simulation by You et al. explicitly revealed the transport channel of photogenerated charge carriers at the g-C_3_N_4_/water interface [33]. Subsequently, the influence of boron/oxygen codoping on the graphitic carbon nitride systems was investigated by Yang et al. [34].

Although a series of studies on heptazine as a photocatalyst for the water splitting reaction have been conducted, to the best of our knowledge, an accurate explanation of the detailed mechanism of heptazine–(H_2_O)_4_ in real liquid water is still lacking. Nevertheless, the size of the water cluster presumably impacts the photochemistry reaction rates and branching ratios. Due to the lack of sufficiently accurate ab initio electronic structure methods, we adopted heptazine–(H_2_O)_4_ as a model to investigate how the size of water clusters impacts the photochemistry mechanism. As is well known, the DFT functional method is usually employed to calculate the excited state charge transfer states, but it cannot provide an accurate description of the charge transfer excitations [35,36]. Thus, it is necessary and important to perform computationally expensive calculations with ADC(2) to capture some subtle phenomena that might be overlooked during the photochemical process. What is the mechanism of the EDPT reaction during the entire process for larger heptazine–(H_2_O)_4_ clusters? How will the size of water clusters affect the EDPT reaction process of heptazine–water clusters? Do the excited state decay patterns differ for both pyridine–water clusters and heptazine–water clusters? What is the effect of decay channels and efficiency when the nonrigid pyridine is replaced by rigid heptazine as a photocatalyst for the water splitting reaction? Inspired by the above-described works and to answer these questions at the molecular level, a detailed trajectory surface hopping dynamics simulation of heptazine–(H_2_O)_4_, based on the Landau–Zener formula, was carried out. The work here lays the foundation for a better understanding of the mechanism behind the water splitting process of heptazine–water complexes within an artificial photosynthetic system.

## 2. Results

### 2.1. Ground State Equilibrium Structures and Absorption Spectrum

For the convenience of comparison and to make the results more intuitive, the optimized ground state equilibrium geometries of the heptazine–H_2_O and heptazine–(H_2_O)_4_, both calculated using the MP2 method, are shown in Figure 1a,b, respectively. The length of the hydrogen bond (RNH) between the nitrogen atom of the heptazine and the hydrogen-bonded water decreases significantly from 2.046 Å in the heptazine–H_2_O complex [37] to 1.96 Å in the heptazine–(H_2_O)_4_ complex, indicating a robust stabilization effect from the surrounding water environment as the size of the water cluster increases. The minimum heptazine–H_2_O complex that we optimized has a planar structure [19], with Cs symmetry from a structural chemistry perspective. This contrasts with the out-of-plane distorted structure predicted by the DFT calculations [38,39], where only one hydrogen bond is formed, facilitating hydrogen transfer from water to the chromophore. This structure is slightly different from the double hydrogen bond geometry reported in a recently published paper, which was optimized using the TDDFT method at the ωB97XD functional level along with the 6-31G(d,p) basis set [26].

For the heptazine–(H_2_O)_4_ complex shown in Figure 1b, the structure of the optimized minimum geometry that we obtained is nonplanar, resulting from the formation of stronger hydrogen bonds among those four water clusters. The Cartesian coordinates of the optimized heptazine–(H_2_O)_4_ complex can be obtained from the Appendix A. This is completely different to the structure that characterizes five water clusters forming a parallel hydrogen-bonded planar cluster above the heptazine chromophore, as calculated by Ullah et al. [26]. To better understand the characteristics of heptazine–(H_2_O)*_n_* (*n* = 1, 4) complexes, the lowest eight vertically excited state energies and oscillator strengths are shown in Table 1.

As can be seen, the eight lowest vertical electronic excitation spectra of the heptazine–(H_2_O)*_n_* (*n* = 1, 4) complexes are all below 5.0 eV, which is consistent with the fact that water cannot absorb photons in this energy range. The calculated excited energy and oscillator strengths of the heptazine–H_2_O [19,26] and heptazine–(H_2_O)_4_ complexes are very similar. The excitation spectrum consists of the four lowest dark states (S1–S4), the other two higher occupied dark states (S7, S8), and two quasi-degenerate bright states (S5, S6). The first bright state is 1.60 eV and 1.75 eV above the first excited state for the heptazine–H_2_O [19] and heptazine–(H_2_O)_4_ complexes, respectively. The excited energy results—calculated here using the ADC(2) method at cc-pVDZ, mixed with the aug-cc-pVDZ basis set level—are all lower than the corresponding values calculated by Ullah using the TDDFT method at the ωB97XD/6-31g(d,p) level, revealing a distinct red-shift feature [26]. Nevertheless, this is in good agreement with the oscillator strengths, which determine whether a state is bright or dark in both methods.

The systematic overestimation of excitation energies by the TDDFT method (compared with the ADC(2) method) is a known limitation, particularly for charge transfer-like transitions in heptazine–water systems. However, we emphasize that the comparison remains valuable for the following two reasons: First, while the ADC(2) method is more accurate for excited state properties, its computational cost becomes prohibitive for larger clusters or dynamical sampling. The TDDFT method, despite its limitations, offers a practical balance between accuracy and efficiency, provided that its systematic errors are acknowledged. Our discussion explicitly highlights these discrepancies to guide readers in interpreting the data. This comparison lays the groundwork for future studies to refine functional choices or combine methods (e.g., the TDDFT method for geometry optimization and the ADC(2) method for vertical excitations). Another reason is to provide a benchmark against the literature. By comparing our ADC(2) results with both TDDFT and literature data, we identify consistent biases that aid in reconciling computational and experimental spectra. This also demonstrates how structural variations in the size of water clusters affect predictions of quantum yield within the same or different methods, which is critical for mechanistic insights.

The computed absorption spectra of the heptazine–H_2_O and heptazine–(H_2_O)_4_ complexes at the ADC(2)/cc-pVDZ level, as well as the aug-cc-pVDZ level, are shown in Figure 2. The peak of the absorption band of the heptazine–(H_2_O)_4_ complex is 0.03 eV blue-shift compared with the absorption band of the heptazine–H_2_O complex [19]. The maximum value of the calculated absorption spectra of the heptazine–H_2_O complex using the ADC(2) method [19] by Ehrmaier et al. [37] and TDDFT method by Ullah et al. [26] are 4.02 eV and 4.81 eV, respectively. In this study, the maximum calculated value of the absorption spectra of the heptazine–H_2_O and heptazine–(H_2_O)_4_ complexes using the ADC(2) method are 4.21 eV and 4.18 eV, respectively. In contrast, the peak of the absorption band for the heptazine–(H_2_O)_5_ complex based on the TDDFT method, calculated by Ullah et al. [26], is 4.89 eV. All values show a blue-shift compared with the experimental results in toluene, a nonpolar and aprotic solvent (3.39 eV) [25].

### 2.2. Nonadiabatic Dynamics Simulations

Based on the available geometries during hopping events, we analyzed all the population hopping reaction mechanisms of both the heptazine–H_2_O and heptazine–(H_2_O)_4_ complexes. As shown in Figure 3a,b, the fractions of 172 and 204 trajectories corresponding to different non-radiation deactivation mechanisms of heptazine–H_2_O and heptazine–(H_2_O)_4_ were plotted, respectively. Starting from the high energy levels of different excited states, 13 of the 172 trajectories of heptazine–H_2_O (7.6%) decayed to the conical intersection seam between the ground state and the first exited state through an electron-driven proton transfer reaction process within 500 fs of the simulation timescale, while the remaining 159 trajectories (92.4%) remained in different high-energy excited states. Among the 13 trajectories, 9 of them decayed to the ground state in tens of femtoseconds, while 4 of them decayed to the ground state in a few hundreds of femtoseconds. For the heptazine–(H_2_O)_4_ complex, 13 of the 200 trajectories (6.5%) de-excited to the conical intersection between the ground state and the first excited state within 500 fs through an electron-driven proton transfer reaction process, while the other 187 trajectories (93.5%) remained in different energy states until the end of the simulation timescale. Among the 13 trajectories, 8 of them decayed to the ground state in tens of femtoseconds, while 5 decayed to the ground state in a few hundreds of femtoseconds. The quantum yield result is slightly lower than the EDPT reaction probability (9%) that was reported for the heptazine–H_2_O complex using the ADC(2) method on a 100 fs timescale by Huang et al. [27], and it is also lower than the 23% EDPT reaction probability that was reported for the heptazine–H_2_O complex using the TDDFT method after 300 fs in [26]. In contrast, the population probability of the S_1_ state in the heptazine–(H_2_O)_4_ complex is higher than those of the heptazine–H_2_O complex, with 91% (92.4% in our 500 fs case) versus 77% being reported based on the ADC(2) method (used by Huang et al. [27]) and the TDDFT method after 300 fs (used by Ullah et al. [26]). It should be noted that since the ADC(2) method may not accurately describe the state coupling near conical intersections, the predicted yields could be skewed. In our case, the trajectory propagation was artificially terminated when the energy gap between the S_0_ ground state and the S_1_ excited state reached a threshold value of 0.2 eV. As a result, the predicted yields may be underestimated. Similarly, the nonadiabatic transition rates might not be correctly captured, potentially leading to either an over- or under-estimation of decay times. We have acknowledged these possible discrepancies in this work and have emphasized the need for further validation using more advanced methods, such as multi-reference methods (e.g., CASSCF, MRCI) and more detailed nonadiabatic coupling calculations. Additionally, we note that future work could involve comparing our results with those obtained from higher-level methods to further assess the reliability of our findings.

Unexpectedly, as the size of the water cluster increases in the heptazine–(H_2_O)*_n_* system, the decay quantum yield from the first excited state to the ground state via the EDPT reaction process decreases slightly. This may be due to competition between the two hydrogen bonds that are formed between the main nitrogen atom of the heptazine and the corresponding hydrogen-bonded hydrogen atom of the water molecules as the number of water molecules increases, as shown in Figure 1. However, the calculations provide convincing evidence that the quantum yield of the EDPT reaction is somehow improved during the deactivation of hopping events when using the heptazine chromophore instead of the pyridine chromophore [24] as a catalyst, demonstrating the superior catalytic properties of heptazine for efficient solar water splitting. It is important to emphasize that, compared with the hexatomic structure in the pyridine–water cluster, the decay via the well-known ring-puckering pathway is almost suppressed in the heptazine chromophore. This is due to the formation of a rigid structure, caused by the assembly of three hexatomic rings in the heptazine, which, from a competitive perspective, favors the EDPT reaction process. Moreover, another interesting phenomenon is the periodic decay from high-energy excited states to the ground state through the conical intersection seam during the EDPT reaction process in the heptazine–(H_2_O)_4_ complex. The dynamics simulations reveal that the rapid hydrogen detachment from hydrogen-bonded water to the heptazine chromophore occurs in two distinct time patterns in the heptazine–(H_2_O)_4_ complex: a fast decay, where the populations decay within tens of femtoseconds, and a slow decay, where the trajectories decay in between 200 and 300 femtoseconds.

The calculated time-dependence populations of the electronic ground state, the seven lowest adiabatic excited electronic ground state, and the seven and eight lowest adiabatic excited electronic states for the heptazine–H_2_O and heptazine–(H_2_O)_4_ clusters are shown in Figure 4a,b, respectively. It is evident that within the first 50 fs, a sharp decay occurs in the higher excited states, leading to a corresponding increase in the population probability of the lower states. At 50 fs, the population probabilities of the two lowest excited states (S_1_ and S_2_) both reach their maximum value of 0.4. Following this, a sustained de-excitation process occurs, where the population of the S_2_ excited state sequentially decays to the S_1_ excited state and then to the S_0_ ground state within the next 50 fs. As we can see in Figure 4a,b, the population of the S_1_ excited state reaches approximately 0.8 at 100 fs. After this point, the population evolves slowly in the first excited state, gradually decaying to the ground state. Notably, the probability of the ground state population remains below 0.1 until the end of the simulation time at 500 fs. This result is consistent with the statistical findings shown in Figure 3 but differs from the 0.23 value that was calculated using the TDDFT method by Ullah et al. [26]. This discrepancy can be attributed to differences in the accuracy of the computational methods used.

To gain a deeper understanding of the EDPT reaction mechanism in the heptazine–(H_2_O)_4_ cluster and explore the differences between the fast and slow decay processes mentioned above, we analyzed two representative trajectories of heptazine–(H_2_O)_4_, each undergoing either fast or slow decay via the conical intersection channel. A description of a more detailed excited state decay mechanism for a single heptazine–H_2_O complex can be found in Huang’s published work [27]. As shown in Figure 5, the potential time-dependence energies of the ground state and the eight lowest excited states, the significant N-H bond length between heptazine and hydrogen-bonded water, the O-H bond length of the intramolecular hydrogen-bonded water, and various orbital energies were all calculated for a representative fast decay trajectory.

When the heptazine–(H_2_O)_4_ complex is excited to the high-energy adiabatic S_6_ state upon absorbing a photon, the molecular complex remains in the excited state until it rapidly transitions to the even higher S_7_ state within just 1 fs, as shown in Figure 5a. This is followed by weak oscillations between the S_6_ and S_8_ states due to the near-degeneracy among these high-energy states. Notably, a downward transition from the S_6_ state to the S_5_ state occurs at approximately 10 fs, after which the population continues evolving at a relatively lower energy level until reaching the conical intersection at 14 fs. This marks a crucial turning point for the hydrogen transfer reaction from water to the chromophore. To further analyze this process, we calculated the time-dependent evolution of the significant N-H bond length between heptazine and the hydrogen-bonded water, as well as the O-H bond length within the intramolecular hydrogen-bonded water, over the first 14 fs. As shown in Figure 5b, following excitation, the N-H bond length initially increases from its optimized ground state value of 1.96 Å to 2.09 Å at 1.5 fs, before rapidly decreasing to 1.47 Å at 10 fs. This decrease signifies that the hydrogen has moved to a central position between the water molecule and chromophore at high speed. Subsequently, the N-H bond length continues decreasing in a free oscillatory mode, reaching 1.15 Å at 14 fs, at which point it hits the conical intersection seam, forming the heptazinyl radical—an essential intermediate in the water splitting process. Simultaneously, the O-H bond length undergoes a sharp decrease from its initial optimized value of 0.95 Å to 0.82 Å at 1.5 fs, as indicated by the red dashed line in Figure 5b. It then increases synchronously with the N-H bond length, reaching 1.47 Å at 10 fs—matching the N-H bond length at that moment—before continuing to increase steadily to 1.69 Å by the end of the simulation. This final increase reveals the formation of another important product: the hydroxyl radical.

In addition, to gain further insight into the EDPT process from water to the heptazine chromophore, the time-dependent orbital energy and its characteristics are presented at each timestep. As shown in Figure 5c, after vertical excitation, the system initially resides in the locally excited ^1^ππ* state in the Frank–Condon region at the start time (0 fs). It is clear from this representation that the HOMO is predominantly located on the peripheral nitrogen atoms of the heptazine structure, distinct from its LUMO characteristic reported by Ehrmaier et al. [19], where their calculations on pure heptazine and its derivatives showed the LUMO to be predominantly localized on the central nitrogen atom and the six peripheral carbon atoms [28]. The system then rapidly transitions to a degenerate n orbital of heptazine and a p_xy_ orbital of hydrogen-bonded water molecules at 1 fs. After 3.5 fs of evolution, the system stabilizes in a p_z_ orbital. During the subsequent transfer process, it exhibits slight oscillations between the n/p_xy_ and p_z_ orbitals due to the complete degeneracy of the energy levels between these two charge transfer states. The representative orbital characters of p_z_ and p_x/y_ are shown at 6 fs and 8 fs, respectively, at the bottom of the figure.

In Figure 6, the time-dependent potential energies of the ground state and the lowest three excited states, as well as the significant N-H bond length between the heptazine and hydrogen-bonded water, the O-H bond length in the intramolecular hydrogen-bonded water, and different orbital energies, are presented for a typical slow decay trajectory. Starting from the first excited S_1_ state, the population evolves on the same potential energy surface until it reaches the corresponding degenerate S_2_ state at 261.5 fs, as shown in Figure 6a with red dots. After this, the S_2_ state potential energy surface crosses the increasing S_0_ ground state at 263.5 fs, and the population reaches the conical intersection at this specific moment.

Similarly, the N-H bond length between the heptazine chromophore and the nearest water molecule weakens significantly after photon absorption, increasing from its initial 1.60 Å to 1.85 Å during the first 256.5 fs, and then decreases directly to 1.11 Å during the remaining simulation time, as indicated in Figure 6b with the black solid line. Meanwhile, the O-H bond length, which links the water molecule that is involved in the intended splitting, shows a corresponding decreasing trend during the first 265.5 fs, from 1.16 Å to 0.79 Å, followed by a sudden increase to 1.55 Å until 263.5 fs, as shown by the red dashed line in Figure 6b. It is important to note that the N-H bond length equals the O-H bond length at 261.5 fs, which aligns with the hopping event time discussed earlier, marking a critical EDPT turning point during the entire process.

From the time-dependent orbital energy results shown in Figure 6c, we can observe that the excited population, starting from the π orbital of the localized heptazine ring, prefers to maintain this characteristic for approximately 261.5 fs, eventually transitioning to the n/p_xy_, as can be seen in the figure at 258 fs and 262 fs, respectively. In comparison to the fast decay case, the slow decay shows a lower excited state energy at the beginning. According to the Heisenberg uncertainty principle, the higher the energy of a state is, the denser the energy levels are, which increases the likelihood of degeneracy between high-energy states. This provides a higher probability for the population to hop to a new excited state, thus favoring the EDPT process. As a result, the N-H bond is more likely to elongate in the opposite direction for a longer period (265.2 fs vs. 10 fs) compared with the fast decay case, providing direct evidence for the delayed EDPT process in the heptazine–water complex. Due to the relatively large gap between the S_1_ and S_2_ states during the first 261.5 fs of the slow decay case, the orbital character does not change during this period, preventing the charge transfer state from crossing with the initial localized excited state and hindering the EDPT event in this scenario.

The analysis of both the fast and slow decay cases provides valuable theoretical insight into the water splitting mechanism. Specifically, to improve the quantum yield and achieve effective water splitting products in the heptazine–water complex, exciting the complex to higher energy levels would be a better strategy to shorten the average lifetime of the excited states. This approach paves the way for further research into the water splitting capabilities of heptazine–water clusters.

## 3. Materials and Methods

The ground state equilibrium geometries of the heptazine–(H_2_O)*_n_* (*n* = 1, 4) complexes were optimized using the second-order Møller–Plesset (MP2) method [40]. Dunning’s augmented correlation-consistent split-valence double-ζ basis set with polarization functions (aug-cc-pVDZ) was applied to the directly hydrogen-bonded water and the nearest potentially valuable double-N atoms, while the remaining atoms were optimized with a correlation-consistent split-valence double-ζ basis set with polarization functions (cc-pVDZ) [41]. This method has been shown to strike a balance between appropriate energy profiles and computational expense. The augmented basis set is essential for the accurate description of ^2^πσ^*^ excited states, which better describe the photochemistry of heteroaromatic systems with acidic groups [42]. Excited state energy, oscillator strength, and energy profile computations were performed using the second-order algebraic diagrammatic construction (ADC(2)) method [43], which has been shown to provide a stable and accurate description of the energy near potential energy surfaces of excited states with conical intersections. All of these electronic structure calculations were carried out using the TURBOMOLE v6.3.1 program package [44]. To further validate the accuracy of the ADC(2) method for the heptazine–water system, previous benchmarking studies provide strong evidence that the results obtained using this method are in good agreement with high-level ab initio CASPT2 calculations [10,24,26,37].

Based on the optimized geometry that we obtained, a set of 2000 initial conditions with different geometries and velocities were extracted from the harmonic oscillator thermal Wigner distribution function. For these configurations, the eight lowest-lying vertically excited energies and corresponding oscillator strengths were calculated for the heptazine–(H_2_O)_4_ complexes. By summing the oscillator strengths over vertically excited energies between the ground state and other excited states—computed at the specific geometry, i.e., using the nuclear ensemble method proposed by Barbatti et al. [45]—the maximum absorption spectra of the heptazine–(H_2_O)_4_ complex were simulated. A width of 0.05 eV, broadened with a Lorentzian function, was applied to the spectra to account for spectral broadening.

In order to explore the effect of the number of surrounding water molecules on the EDPT process from water to the chromophore in the heptazine–(H_2_O)_4_ complex, nonadiabatic dynamics simulations of 204 individual trajectories, starting from different excited states, were carried out using a trajectory surface hopping method based on the Landau–Zener formula. The corresponding algorithm has been described in related work [45,46,47], and it has been proven to provide reliable calculation results for the EDPT process from water to pyridine in our previous work [24]. The total simulation time for the molecular dynamics simulations was set to 500 fs for both systems, with a time step of 0.5 fs to propagate the nuclear motion using the velocity Verlet algorithm. One should note that since the ADC(2) method is a computationally efficient single-reference propagator method, the calculated excited state energy profiles may be inaccurate in the vicinity of a conical intersection. Therefore, to ensure the scientific rigor and accuracy of our calculations, the nonadiabatic dynamics were artificially terminated once the population reached a region near the conical intersection. Specifically, we terminated the trajectory propagation when the energy gap between the S_0_ ground state and S_1_ excited state reached a threshold value of 0.2 eV. As pointed out above, the ADC(2) method may not adequately capture the electronic structure and dynamics in these regions, especially regarding the coupling between electronic states and nonadiabatic effects. This limitation may impact key results, such as the reaction yields and decay times, but it does not affect our understanding and exploration of the basic mechanism. We recommend that future studies explore more accurate methods to improve the understanding of nonadiabatic dynamics at conical intersections.

## 4. Conclusions

Using trajectory-based Landau–Zener surface hopping molecular dynamics, combined with ADC(2) electronic structure calculations and a mixed cc-pVDZ/aug-cc-pVDZ basis set, we investigated the electron-driven proton transfer (EDPT) process from water to the heptazine chromophore in a heptazine–(H_2_O)_4_ complex in detail. By simulating 204 trajectory calculations for the heptazine–(H_2_O)_4_ complex, we confirmed the occurrence of the EDPT process in heptazine–(H_2_O)_4_ complexes, leading to the formation of a heptazinyl and OH biradical, as detected in our simulations. The calculated quantum yield of EDPT for a heptazine–(H_2_O)_4_ complex is 6.5%, suggesting that the addition of more water molecules does not substantially enhance the hydrogen transfer process. Furthermore, we identified two distinct decay patterns through mathematical statistical analysis of EDPT events in heptazine–(H_2_O)*_n_* (*n* = 1, 4) complexes: (1) a fast decay event, where populations transition from the excited state to the ground state within a few tens of femtoseconds, and (2) a slow decay event, occurring over a few hundred femtoseconds.

To gain a deeper insight into this phenomenon, we selected representative cases of both fast and slow decay events in the heptazine–(H_2_O)_4_ for further mechanistic investigation. Direct monitoring of the key N-H bond length between heptazine and its bonded water molecule, as well as the O-H bond length within the hydrogen-bonded water molecule, provided strong theoretical evidence supporting the EDPT reaction. A critical finding of this study is that the moment when the N-H bond length equals that of the O-H bond marks a critical turning point in the EDPT reaction, coinciding precisely with the population hopping event and the change in orbital character, governing the overall macroscopic picture of the EDPT process.

The presence of fast and slow decay usually indicates multiple relaxation mechanisms, different molecular states, or competing processes. Time-resolved spectroscopy may help distinguish these contributions. The observation of fast and slow decay components in this EDPT process in heptazine–(H_2_O)*_n_* (*n* = 1, 4) systems probably arises from different relaxation pathways or competing processes. Possible reasons for this include the following: It may be caused by the involvement of a triplet state. Usually, decay is fast in fluorescence due to single-state relaxation, while in phosphorescence, decay is slow due to forbidden triplet-to-singlet transitions. If both processes occur, the system exhibits fast and slow decay components. In our case, however, heptazine (1,3,4,6,7,9,9b-heptaazaphenalene or tri-*s*-triazine), as proven by Ehrmaier et al., exhibits an inverted S1/T1 energy gap (∆ST ≈ −0.25 eV) [28]. We expect that scientists with a strong theoretical foundation will be able to offer a better explanation for this phenomenon.

These insights establish a solid foundation for further optimizing the quantum yield of water splitting reactions, particularly when using heptazine-based molecular systems as photocatalyst candidates.

## Figures and Tables

**Figure 1 ijms-26-04549-f001:**
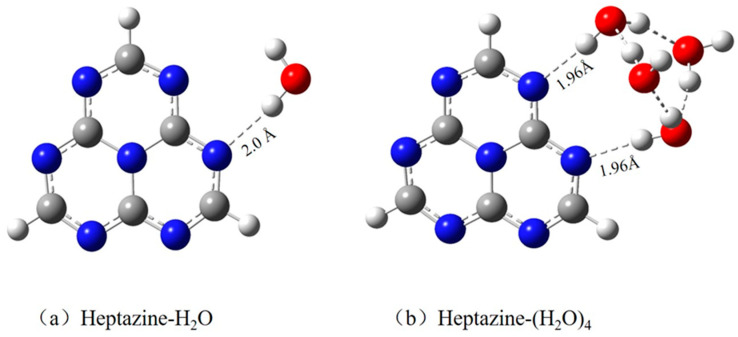
Ground state equilibrium geometries of (**a**) heptazine–H_2_O cluster and (**b**) heptazine–(H_2_O)_4_ clusters, calculated using the MP2 method with cc-pVDZ mixed with the aug-cc-pVDZ basis set.

**Figure 2 ijms-26-04549-f002:**
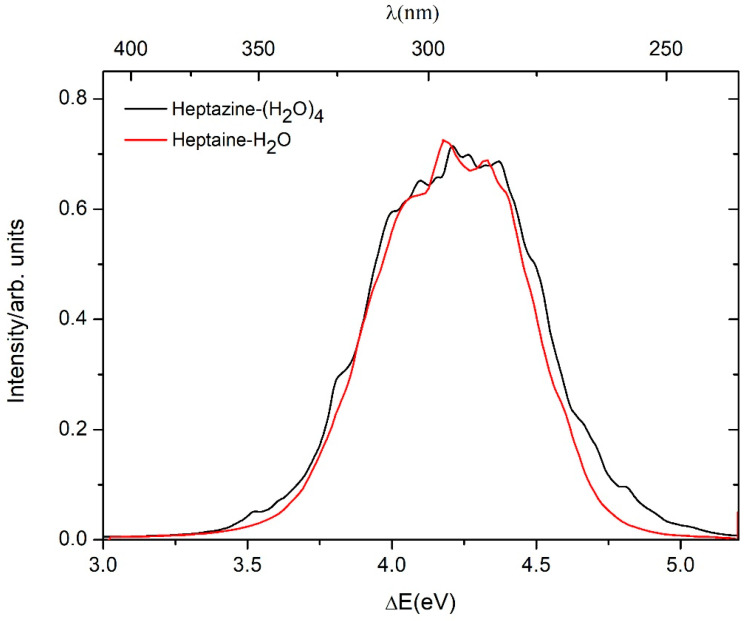
Absorption spectra of the heptazine–H_2_O (red line) and heptazine–(H_2_O)_4_ (black line), simulated at the ADC(2)/cc-pVDZ level, as well as the aug-cc-pVDZ level.

**Figure 3 ijms-26-04549-f003:**
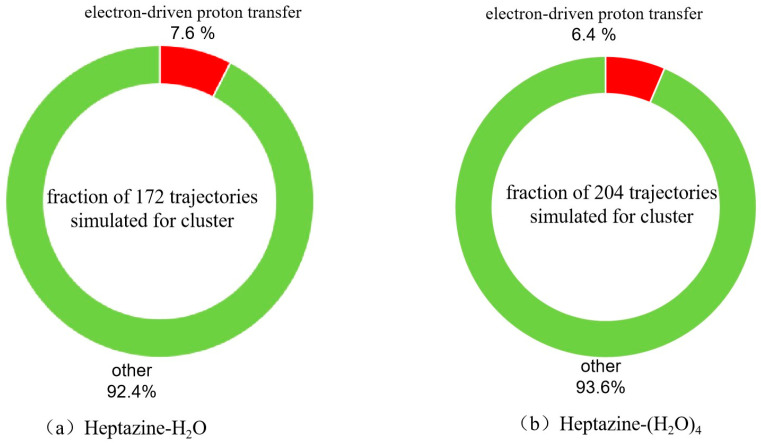
Branching ratios of different excited state deactivation channels for (**a**) heptazine–H_2_O and (**b**) heptazine–(H_2_O)_4_ with a simulation timescale of 500 fs.

**Figure 4 ijms-26-04549-f004:**
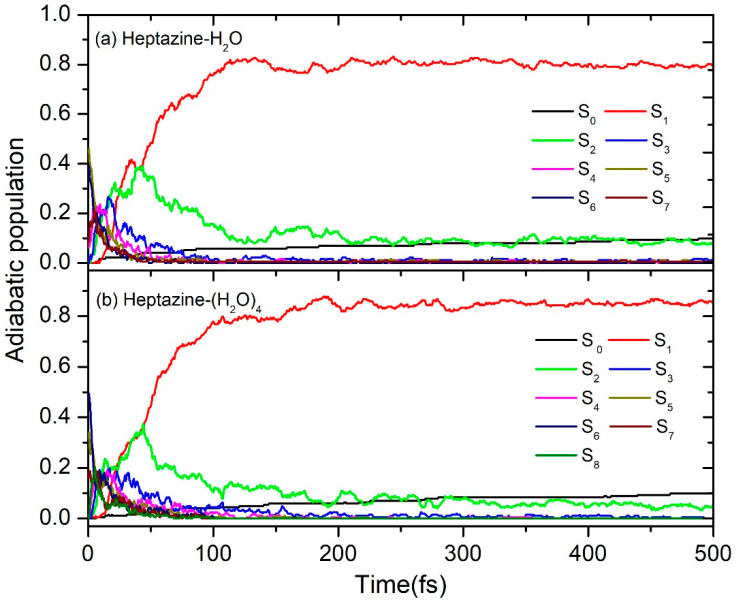
Population probabilities of the electronic ground state and (**a**) the seven lowest adiabatic excited electronic states for heptazine–H_2_O and (**b**) the eight lowest adiabatic excited electronic states for the heptazine–(H_2_O)_4_ clusters.

**Figure 5 ijms-26-04549-f005:**
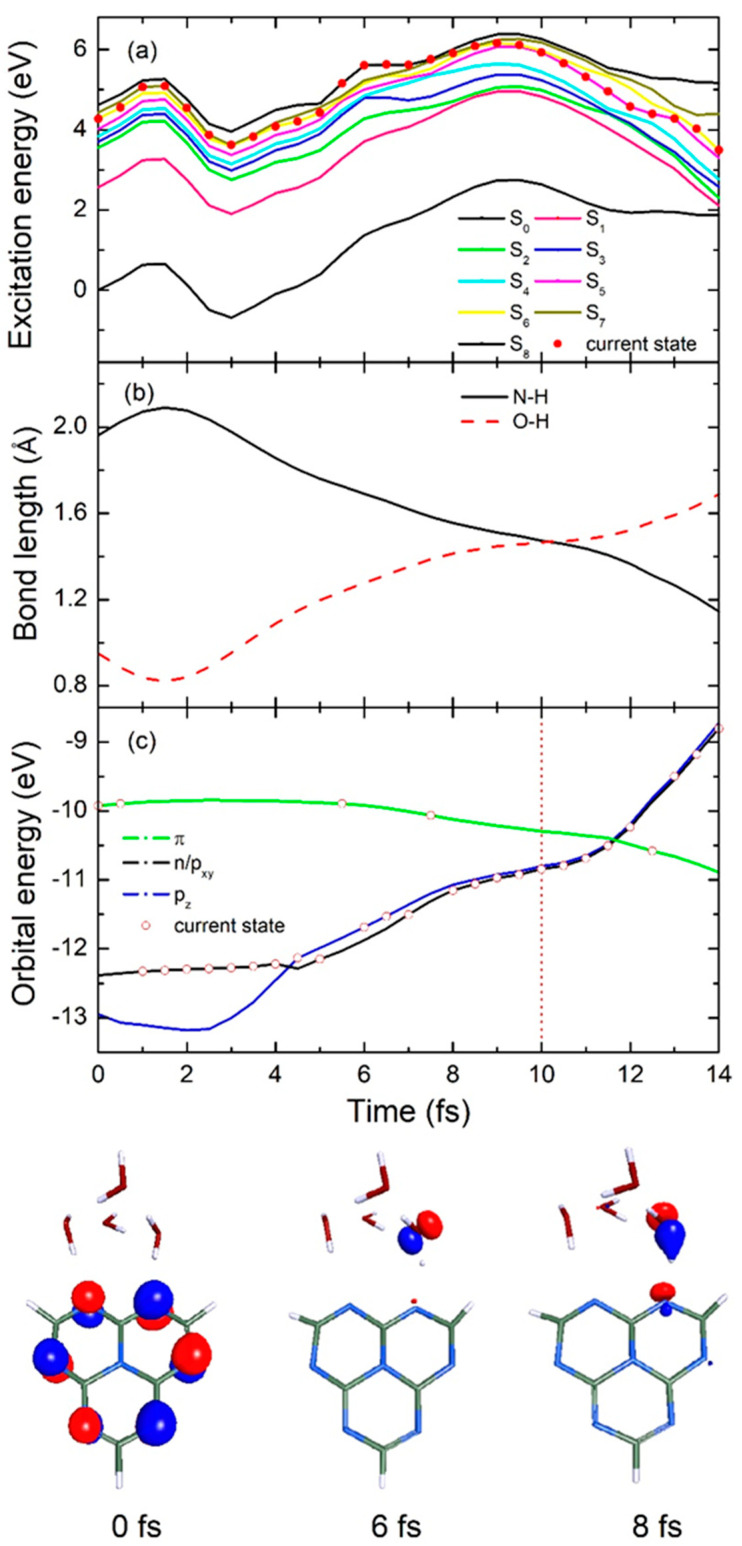
Time−dependence of (**a**) 8 lowest excited state and ground state energies, (**b**) significant N-H bond length between heptazine and hydrogen−bonded water and O-H bond length on intramolecular hydrogen-bonded water, and (**c**) different orbital energies in a typical single representative fast decay trajectory for heptazine–(H_2_O)_4_ complex. Bottom half of figure shows selected orbital characteristic over three different specific simulation times.

**Figure 6 ijms-26-04549-f006:**
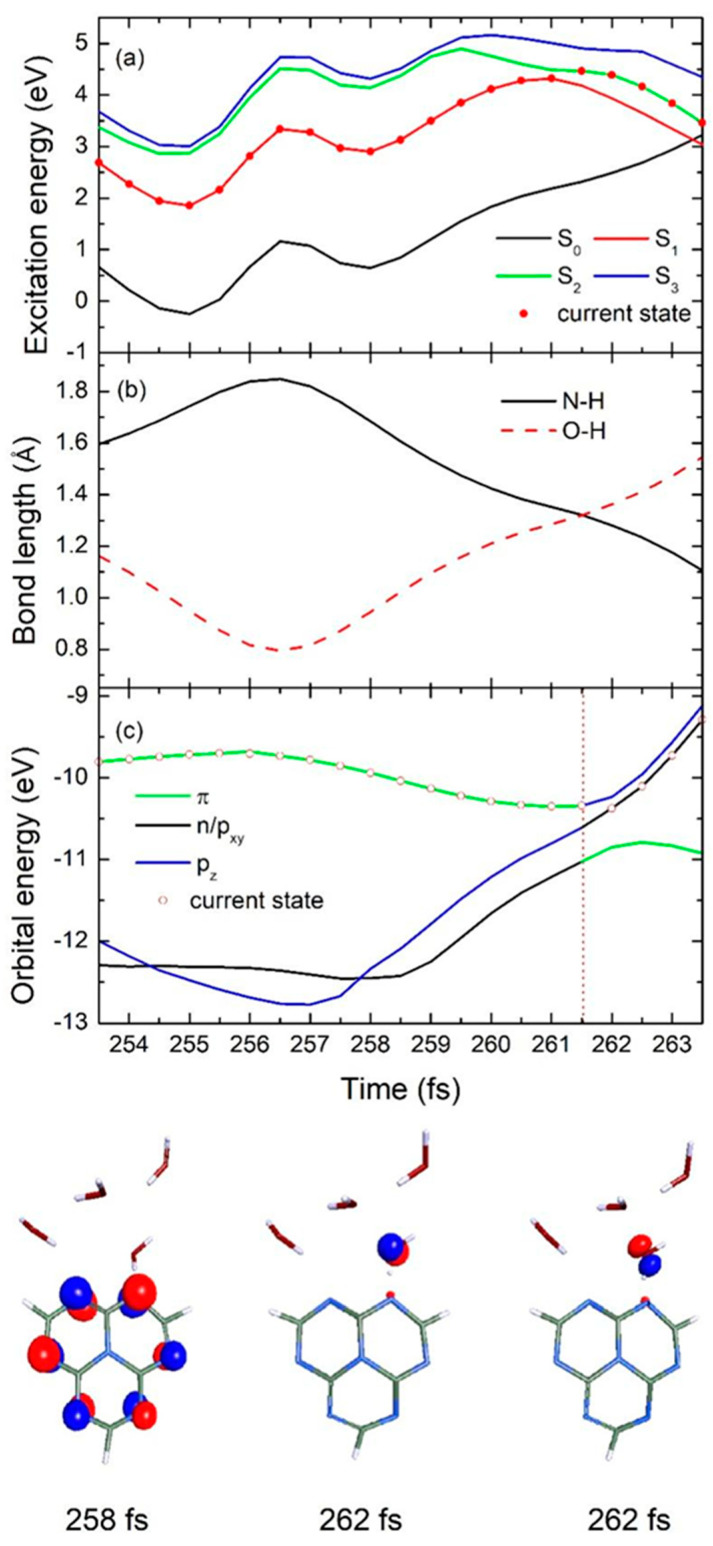
Time–dependence of (**a**) 3 lowest excited state and ground state energies, (**b**) significant N-H bond length between heptazine and hydrogen–bonded water and O-H bond length of intramolecular hydrogen-bonded water, and (**c**) different orbital energies in typical single representative slow decay trajectory for heptazine–(H_2_O)_4_ complex. Bottom half of figure shows selected orbital characteristics at three different specific simulation times.

**Table 1 ijms-26-04549-t001:** The eight lowest vertical electronic excitation energies (in eV) and oscillator strengths (f) of the heptazine–H_2_O and heptazine–(H_2_O)_4_ complexes.

Heptazine–H_2_O			Heptazine–(H_2_O)_4_		
State	Energy	f	State	Energy	f
S_1_	2.60	0.00	S_1_	2.63	0.00
S_2_	3.72	0.00	S_2_	3.74	0.00
S_3_	3.80	0.00	S_3_	3.85	0.00
S_4_	3.89	0.00	S_4_	4.00	0.00
S_5_	4.20	0.26	S_5_	4.38	0.25
S_6_	4.22	0.25	S_6_	4.40	0.30
S_7_	4.75	0.00	S_7_	4.83	0.00
S_8_	4.80	0.00	S_8_	4.97	0.00

## Data Availability

The original contributions presented in this study are included in the article/Appendix A. Further inquiries can be directed to the corresponding authors.

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
