# Peer review of "Nonadiabatic Surface Hopping Dynamics of Photocatalytic Water Splitting Process with Heptazine–(H2O)4 Chromophore"

_ijms, 2025, doi:10.3390/ijms26104549_

Round 1

Reviewer 1 Report

Comments and Suggestions for Authors

This manuscript provided an insight to the photochemical mechanisms of heptazine-based polymeric carbon nitride materials using the Landau-Zener surface hopping approach. The majority of data supports the argument raised in the manuscript. I would recommend a major revision before accepting the paper. The questions/suggestions I have are listed below:

Author Response

Please see the attachment, thank you!

Reviewer 2 Report

Comments and Suggestions for Authors

Nonadiabatic surface-hopping dynamics of photocatalytic water splitting process with heptazine-(H2O)4 chromophore

This manuscript provides computational study of the electron-driven proton transfer (EDPT) process in a heptazine-(Hâ‚‚O)â‚„ complex using ADC(2)-level surface hopping dynamics. The authors analyze nonadiabatic decay mechanisms, quantum yields, and identify fast vs. slow decay patterns.

The manuscript is theoretical insights into photocatalytic water splitting.  However, there are some areas that require clarification and additional explanation as listed in the comments below. I would accept this manuscript with major revision.

  1. The manuscript contains numerous grammatical errors, awkward phrasings and long sentences. Please read throughout the manuscript to correct the sentences.
  2. Author claims that "periodic decay behavior presents a promising strategy for enhancing water-splitting efficiency under sunlight" is speculated but no practical or reference provided. Please address this.
  3. As authors acknowledge ADC(2)’s limitations near conical intersections, there is little critical discussion of how this affects key findings such as yield values, decay times. Please address this in manuscript.
  4. Please correct the figures 3–6, as they are not fully explained or integrated into the flow of the results. Need more in depth discussion.

Author Response

Please see the attachment, thank you!

Round 2

Reviewer 2 Report

Comments and Suggestions for Authors

I am okay with current version.